# Diversity of soil bacterial communities in response to fonio (*Digitaria exilis* Stapf) genotypes and pedoclimatic conditions in Benin

Tania L. I. Akponikpè [1,2]*, Elvire L. Sossa[3], Enoch G. Achigan-Dako[2], Guillaume L. Amadji[3], Séverine Piutti[1]

1 Université de Lorraine, INRAE, LAE, Nancy, France, 2 Genetics, Biotechnology and Seed Science Unit (GBioS), Laboratory of Plant Production, Physiology and Plant Breeding (PAGEV), Department of Plant Sciences, Faculty of Agronomic Sciences, University of Abomey-Calavi, Abomey Calavi, Republic of Benin, 3 Unité de Recherche Gestion Durable de la Fertilité des Sols, Laboratoire des Sciences du Sol, Faculté des Sciences Agronomiques (FSA), Université d'Abomey-Calavi (UAC), Abomey Calavi, Bénin

* lorraine.akponikpe@univ-lorraine.fr

## Abstract

In sub-Saharan regions, soil fertility is a major concern for plant productivity, influenced by physical, chemical, and biological components. Among biological properties, the recruitment of soil microbial communities by plant roots is influenced by both physico-chemical soil properties and plant characteristics, dependent on species or genotypes. Here, rhizosphere bacterial communities associated with five fonio genotypes cultivated under three pedoclimatic conditions were investigated. Rhizosphere soils were collected for high-throughput 16S rRNA gene sequencing to characterize soil bacterial diversity. Additional parameters were assessed to classify soil fertility of the three pedoclimatic conditions and to evaluate relationships between the bacterial community's composition and soil fertility variables. Principal Component Analysis revealed a clear effect of pedoclimatic condition, whereas genotype had no significant impact on soil chemical properties or enzyme activities. Overall, soils were low in fertility, with Boukoumbe soil standing out for its higher chemical values and enzyme activities. For example, Boukoumbe reached 1.48% organic carbon, compared to 0.61% in Gogounou and 0.36% in Ina. Similarly, total nitrogen and available phosphorus were also higher in Boukoumbe. Regarding bacterial community, there is no impact of pedoclimatic condition and genotype on their richness and diversity. However, Bray-Curtis index revealed a significant difference in bacterial community structure among pedoclimatic conditions, but not among fonio genotypes. This suggests, in rhizosphere, bacterial community structure is more modulated by soil properties than crop genotypes. Proteobacteria and Bacteroidota were most abundant phyla, varying significantly across pedoclimatic conditions. Moraxellaceae and Oxalobacteraceae bacteria were most abundant families within Proteobacteria, while Chitinophagaceae and Weeksellaceae dominated in Bacteroidota. Our study

**Data availability statement:** All relevant data are publicly available from the Dataverse repository (https://doi.org/10.57745/FQAJBU).

**Funding:** This research was funded by French National Research Agency through the PEA-BIOVALOR (ANR-21-PEA2-0006) Biovalor project, within the framework of Tania L. I. Akponikpè's PhD. The funders had no role in study design, data collection and analysis, decision to publish, or preparation of the manuscript.

**Competing interests:** The authors have declared that no competing interests exist.

highlighted the significant roles of soil pH, as well as sulfate and nitrate content, in shaping bacterial communities. These findings offer valuable insights into the bacterial communities associated with fonio and their key drivers.

## Introduction

Soil, the foundation of agricultural activities, is the primary source of nutrition for plants, animals, and humans through its fertility [1]. Fertile soils provide essential nutrients for crop plant growth, support a diverse and active biotic community, exhibit a typical soil structure, and allow for an undisturbed decomposition [2]. Soil fertility results from its physical, chemical, and biological properties [3], which are in interaction. To evaluate the potential productivity of sub-Saharan soils, several properties are required, including pH, texture, soil organic carbon, bulk density, depth, stone content, and water-holding capacity [4]. Plant productivity also depends on microbial abundance and functions, as bacteria and fungi are key actors of nutrient availability and plant health. While extensive research has focused on the physical and chemical characteristics of soil fertility in sub-Saharan Africa [4–6], little attention has been devoted to microbial indicators in the soil. Microorganisms, the most abundant living entities in the soil, play a key role in nutrient availability and root interception through their involvement in nutrient cycling, organic matter mineralization, plant growth hormones production and pathogenic control [7–10]. In recent years, the soil microbiome has gained recognition for its critical role in plant growth, protection, and soil nutrient dynamics, making it a focal point of agricultural research and highlighting the need to integrate biological indicators into soil fertility assessments [1,11–13].

The structure and diversity of soil microbial communities are shaped by various factors, acting as ecological filters. These include abiotic factors such as soil properties, climatic conditions and biotic factors like land use, plant species, and microbial interactions [14–22]. Soil properties such as texture, pH, organic matter content and nutrient availability, and plant species provide a unique microenvironment that affects the establishment and activity of microbial communities. For instance, several *Pseudomonas* strains have shown a particular preference for phosphorus-deficient, saline or low-temperature agricultural soils and can positively influence enzyme activity such as phosphatase [23–25], helping plants in stress management. Abiotic factors such as pH, temperature and nutrient concentrations also strongly influence the production of microbial enzymes, which are considered to be key indicators of microbial functioning. Mainly produced by microorganisms, these enzymes, known for their catalytic efficiency, are essential to microbial metabolism and reflect both the quality of organic matter and the metabolic activity of microbial communities [26,27]. Likewise, plant species play a decisive role in structuring soil microbial communities. It has been shown that the introduction of different plant species contributes to variations in bacterial community composition and helps to restore soil health [14,16]. Beyond species, the effects of plant cultivar or genotype have also been widely reported, both in the rhizosphere and in the phyllosphere [16,28–31]. For example, wheat genotypes that differ in plant height show distinct bacterial communities in their rhizosphere [32].

Furthermore, due to their genetic variability, genotypes may exhibit different functional characteristics that could potentially explain the release of specific root metabolite profiles, thereby creating distinct microhabitats leading to differentiated microbial assemblages. Considering the root environment, rhizodeposition along with plant phenology affect the properties of the rhizosphere, which is a dynamic zone where microorganisms and plant roots interact mostly [33,34]. While genotype effects have been reported in crops like maize, rice, and millets [28,29,34], several studies suggest that local soil conditions may exert a stronger influence than plant genotype [30].

However, despite advances in understanding plant-microorganisms interaction across various crop species, knowledge remains limited for certain neglected and underutilized crops, notably Fonio (*Digitaria exilis* Stapf), leaving a significant research gap to be addressed. Fonio, a drought-tolerant cereal endemic to West Africa, plays a significant role in food and nutritional security, particularly in rural areas [35,36]. Naturally gluten-free and rich in methionine and fiber, fonio is predominantly cultivated on degraded, low-fertility soils across the Sudanian–Guinean zones [37–39]. In Benin, northern fonio-growing regions are characterized by shallow, stony, and organic matter–depleted soils, conditions that usually constrain microbial abundance and activity [40]. Despite this, fonio remains resilient, suggesting the possible existence of beneficial plant–microbe interactions that enhance its adaptation to poor soils. Similar plant–microbiota associations have been observed in other small-grain cereals, such as millets, where microbial partners contribute to the plant growth [41]. However, while microbial communities in other millets have received attention, very few studies have explored those associated with fonio genotypes [17,42].

In this study, we investigated the richness and composition of rhizosphere bacterial communities associated with five fonio genotypes across three pedoclimatic conditions in northern Benin using 16S rRNA gene sequencing and predicted their functions. Soil physicochemical indicators and enzyme activities related to biogeochemical cycles were measured. We hypothesized that i) the chemical and microbial properties of fonio rhizosphere soils are more strongly dependent on pedoclimatic conditions than on a genotype effect, ii) bacterial community structure may be shaped by both pedoclimatic conditions and plant genotype, iii) the composition of rhizosphere bacterial community could be explained by soil parameters related to nutrient availability, and iv) the bacterial community associated to fonio rhizosphere hosts a diverse functions that could be related to soil health and fertility.

## Materials and methods

### National and local approval

Before embarking on the fieldwork for sampling, the first author secured the approval of the academic committee of the École Doctorale des Sciences Agronomiques et de l'Eau (EDSAE) of the Faculty of Agricultural Sciences (FSA) of the University of Abomey-Calavi (UAC). The ethics committee that waived the study was the first author's thesis committee at the University of Abomey-Calavi comprising professors Enoch G. Achigan-Dako, Adam Ahanchede, Guillaume Amadji. Prior to rhizosphere soil sampling, the objectives of the research project were presented to the local authorities of each village under investigation. The informed verbal consent of all local authorities was obtained before sample collection.

### Study area and rhizosphere soil sampling method

Three municipalities of Benin were studied: Ina (North West; 9.96470°N; 2.70775°E), Boukoumbe (North West; 10.2219°N; 1.1175°E), and Gogounou (North East; 10.7438°N; 2.8080°E). Boukoumbe and Gogounou have a Sudanian climate, with rainfall between 950 and 1,200 mm, while Ina has a Sudano-Guinean climate, with rainfall between 1,200 mm and 1,300 mm. The soil type in this area is ferruginous, characterized by trees, shrubby Savannah and clear forests [43,44]. Table 1 presents the main physico-chemical properties of the studied soil in the three pedoclimatic conditions.

At each pedoclimatic condition, field trials were conducted using a randomized complete block design with three replicates and eleven modalities of a single studied factor (i.e., fonio genotypes). Five fonio genotypes selected as ideal

**Table 1. Some characteristics of soils in the pedoclimatic conditions studied.**

| Parameters | Boukoumbe | Gogounou | Ina |
|---|---|---|---|
| Clay (%) | 11.97 | 9.80 | 8.04 |
| Silt (%) | 28.80 | 20.66 | 17.99 |
| Sand (%) | 58.64 | 68.87 | 73.10 |
| pH (H$_2$O) | 5.19 | 6.19 | 5.24 |
| Organic carbon (%) | 1.14 | 0.85 | 0.59 |

genotypes based on their growth (plant density, plant height, precocity in maturity) [45]: genotypes BEN12, MAL05, MAL14, MAL15, and NIG35 (Table 2) were planted at each site (pedoclimatic condition) to assess their agronomic performance in different environments. Fonio seeds were sown within a 4 m² (2*2 m²) plot. Sowing was carried out in 20-cm-spaced rows with 5 seeds per hole. These experimentations were carried out from June to August 2023 on fonio farmers' fields and at the National Institute for Agricultural Research of Benin (INRAB) research field; permissions were obtained before installation and data collection.

At each site, rhizosphere soil was collected from three plants per genotype per replication, pooled to form one composite sample, representing our sampling unit. Then, three composite samples were collected per site, and 45 samples collected in total (3 sites * 3 replicates * 5 genotypes). At the laboratory (Laboratory of Agronomy and Environment, Nancy-France), the soil was properly separated from roots to have strictly rhizosphere soil that was sieved at 5 mm and stored at 4°C for determination of soil chemical and microbial properties, except for metabarcoding analyses for which soil samples were stored at −40°C.

## DNA extraction and high-throughput sequencing of 16S rRNA gene amplicons

500 mg from each rhizosphere soil was used for environmental DNA extraction performed with the Fast DNA Spin kit for soil (MP Biomedicals, France) following the manufacturer's protocol. The quantity and quality of environmental DNA extracts were then checked by the spectrophotometer and stored at −20°C until further use. DNA was quantified using the Quant-iT dsDNA Assay Kit High Sensitivity (Invitrogen Q33120).

Genoscreen performed DNA sequencing to amplify the V4-V5 regions of the 16S rRNA gene. The procedure used was as follows: (i) preparation of amplicon libraries using Metabiote® solution, limiting amplification bias between samples and including a positive control (artificial bacterial community "ZymoBIOMICS") and negative control (background PCR of the entire library preparation process); (ii) sequencing of amplicon libraries on a single Illumina MiSeq pair-end 600-cycle V2 chemistry cycle; (iii) sorting of sequences per sample and adjustment of the specific primers used to amplify the V4-V5 region of the 16S rRNA gene. The data resulting from this sequencing were available on https://doi.org/10.57745/FQAJBU.

**Table 2. Some characteristics of the genotypes studied.**

| N° | Origin | Genotypes | Maturity cycle (days) | Average yield (kg/ha) | Plant Height (cm) |
|---|---|---|---|---|---|
| 1 | Benin | BEN12 | 105 | 264 | 70 |
| 2 | Mali | MAL05 | 78 | 388 | 80 |
| 3 | Mali | MAL14 | 78 | 478 | 73 |
| 4 | Mali | MAL15 | 96 | 412 | 73 |
| 5 | Niger | NIG35 | 85 | 418 | 77 |

Source: Ibrahim Bio Yerima et al. [46].

## Determination of soil parameters

Several soil parameters were determined. The cation exchange capacity (CEC) and exchangeable bases (Ca$^{2+}$, Mg$^{2+}$, and K$^+$) were extracted from 0.0166 M Co(NH$_3$)$_6$Cl$_3$ at a soil solution ratio of 1:20 (2.5 g,50 ml) and after 1 hour of shaking, according to international ISO standard 23,470 (NF EN ISO 23470:2007). The soil pH was determined in distilled water with a 1:5 soil: water ratio (NF ISO 10390:2005) with a pH meter (Mettler Toledo). The assimilable phosphorus was extracted and measured by spectrometric determination of the soluble part in sodium hydrogen carbonate solution [47]. Total nitrogen (TN) and organic carbon (C_org) were determined by combustion at 900°C with a CHNS analyser (vario MICRO cube, Elementar Analysensysteme GmbH). The hot water carbon (HWC) and nitrogen (HWN) were quantified by determination of their soluble part in water after shaking [48]. The nitrate (NO$_3^-$) and ammonium (NH$_4^+$) contents were measured in an extract soil: KH$_2$PO$_4$ (1:5) filtered on a Whatman filter and 0.2 µm.

The enzyme activities quantified by fluorimetry were: beta-glucosidase (BGLU), arylsulfatase (ARS), acid and alkaline phosphatases (P and Palc), leucine aminopeptidase (LAP), xylosidase (Xylo), and beta-N-acetylglucosaminidase (NAG) according to the method described by Clivot et al. [49]. 4 g of fresh soil was mixed with 50 ml of 0.1 M Tris-HCl buffer pH 8.0, and diluted at a ratio of 1:100. In a microplate, 100 µL of 200 µM substrate (defined according to the enzyme to be measured, Table 3) was then added to 100 µL of the sol solution; four replicates were performed per sample. A standard curve is also produced using 4-methylumbelliferone (MUF) or 7-amino-4-methylcoumarin (AMC) (Table 3) at concentrations of 0 µM, 2.5 µM, 10 µM, and 25 µM; 100 µL of each concentration was added to 100 µL of the soil solution. The mixture was left to incubate for varying lengths of time depending on the enzyme (Table 3). After incubation at 37°C, enzyme activities were measured using fluorimetry by taking a reading at an excitation wavelength of 360 nm and an emission wavelength of 460 nm. Values were then expressed in micromoles per hour per gram of dry equivalent soil (µmol/h/g).

## Bioinformatic and statistical analysis

Two 300 bp reads (R1 and R2) were performed per sample; the "Metabarcoding FROGS 4.1" tool from the FROGS platform was used for sequencing data processing [50] and statistical analysis (based on R 4.5.0 software) [51]. An initial pre-processing step was carried out to gather read pairs, remove adapters, and clean up the sequences.

Then, a clustering step was performed using the sequence similarity method (the SWARM method). The next step was removing clusters considered singletons and chimeras formed during PCR. After this clean-up, amplicon sequence variants (ASVs) were obtained, and ready for taxonomic affiliation. Taxonomic assignments were then made using a Bayesian approach with the SILVA 138.1 Pintail 100 database. The final abundance table was obtained after filtering ASVs with less than 95% of identity and removing of Archaea domain.

Normalization was performed at 11,057 sequences per sample before data analysis. Statistical analysis of bacterial community diversity was achieved using the R software, package « Vegan ». Alpha diversity indices were then estimated: the observed ASV richness, the Chao estimated ASV richness, the Shannon-Wiener diversity index, and the Simpson

Table 3. The enzymes determined from fonio rhizosphere soil samples and the corresponding substrates. 4-methylumbelliferone (MUF), 7-amino-4-methylcoumarin (AMC).

| Enzymes measured | Substrates | Incubation time | Standard used |
|---|---|---|---|
| Arylsulfatase (ARS) | 4-Methylumbelliferyl-sulfate | 6 hours | MUF |
| β -glucosidase (BGLU) | 4-Methylumbelliferyl- β -D-glucopyranoside | 2 hours | MUF |
| Leucine aminopeptidase (LAP) | L-Leucine-7-amido-4-methylcoumarin | 3 hours | AMC |
| β-N-acetylglucosaminidase (NAG) | 4-Methylumbelliferyl-N-acetyl- β -D-glucopyranoside | 6 hours | MUF |
| Phosphatase (P and Palc) | 4-Methylumbelliferyl-phosphate | 2 hours | MUF |
| Xylosidase (xylo) | 4-Methylumbelliferyl-B-D-xylopynanoside | 6 hours | MUF |

diversity index. An analysis of variance (ANOVA) was also performed on the alpha diversity indices to assess the effect of the factors studied (pedoclimatic conditions and genotypes) on ASV richness and abundance with the aov function in R software, and the Tukey test ($P < 0.05$) was used for comparison between significant factors. The Shapiro-Wilk test and Levene's test were used, respectively to check the normality and homogeneity of the alpha diversity data. Beta diversity indices were explored through the Bray-Curtis index; this was analyzed using the NMDS plot (Non-metric Multi-Dimensional Scaling into a 2D representation) according to the pedoclimatic conditions and genotypes studied. Then a PERmutational Multivariate ANalysis of VAriance (PERMANOVA) was performed on the Bray-Curtis index to assess the effect of the factors studied on variations among samples using the adonis function. The Analysis Of Similarities (ANOSIM) was performed using the anosim function in the Vegan package to obtain the strength (statistic R) and the significance to determine the differences between bacterial communities depending on the factors studied. ANOVA and Kruskal-Wallis test were also applied to the bacterial phyla and families. Furthermore, the potential functions of the bacterial communities were inferred from the 16S rRNA gene sequences using the Tax4Fun2 [52], which predicts functional profiles by mapping taxa to the KEGG (Kyoto Encyclopedia of Genes and Genomes) Orthology database for prokaryotes.

A principal component analysis (PCA) was performed in R software for soil chemical and biological parameters, using the prcomp function in the FactoMineR package. Projection plots of individuals on the first 2 principal components were made with the fviz_pca_ind function of the Factoextra package to observe the distribution of individuals according to the variables studied. An ANOVA and Kruskal-Wallis test were applied to soil parameters to prove the statistical differences of the effects identified in PCA. Pearson's correlation analysis using « corrplot », an R package, was then performed to determine the relationships between the soil chemical and biological variables. A first selection of soil variables was made based on their correlation to reduce redundancy; this was followed by a second variable selection using the ordiR2step function in the Vegan package. The significant variables selected were then used to perform a redundancy analysis (RDA) with the rda function. Graphical construction of barplots, heatmaps, and boxplots was carried out using the « Ggplot2 » package.

## Results

### Fertility status of fonio rhizosphere soil

We performed a PCA on the soil chemical properties and enzyme activities to assess the pedoclimatic condition and genotype effect (Fig 1). A total of 73.3% of the variance was explained by the first two dimensions (Fig 1a), respectively 61.8% for PC1 and 11.5% for PC2. The PCA biplot revealed differences in soil by pedoclimatic condition, highlighting their distinct chemical and biological profiles (Figs 1b and 1c). In contrast, genotype points clustered together, indicating that the observed variability in the soil parameters cannot be attributed to genotype differences. Boukoumbe's samples were distributed along the first axis (Dim1), with a large ellipse indicating greater variability at this pedoclimatic condition. The coordinates of Boukoumbe along PC1 were mostly negative, corresponding to higher values of the measured parameters compared to the soils from Gogounou showed positive coordinates along PC1. Gogounou points were more clustered, showing moderate variability along both dimensions and at Ina they were tightly grouped and close to the origin, suggesting minimal variability within this pedoclimatic condition compared to others (Fig 1b). The clear separation of Boukoumbe indicates that this pedoclimatic condition may have distinct soil chemical properties and enzyme activity. Along the second axis PC2, we observed a contrast between Gogounou and Ina, Gogounou points had positive coordinates, whereas those from Ina were negative. This opposition along PC2 could reflect differences in phosphorus and potassium contents between the two zones.

The PCA results were confirmed with the results presented in Table 4. The soil chemical properties were significantly varied among the three pedoclimatic conditions studied. Regarding soil $Ca^{2+}$ content and CEC, Boukoumbe and Gogounou were similar to and higher than Ina at 63% and 52% for $Ca^{2+}$ content and 30% and 21% for CEC, respectively. Soil $K^+$ at Gogounou ($0.1 \pm 0.01$ cmol+/kg) was significantly higher than at Boukoumbe and Ina. $Mg^{2+}$ levels were different for

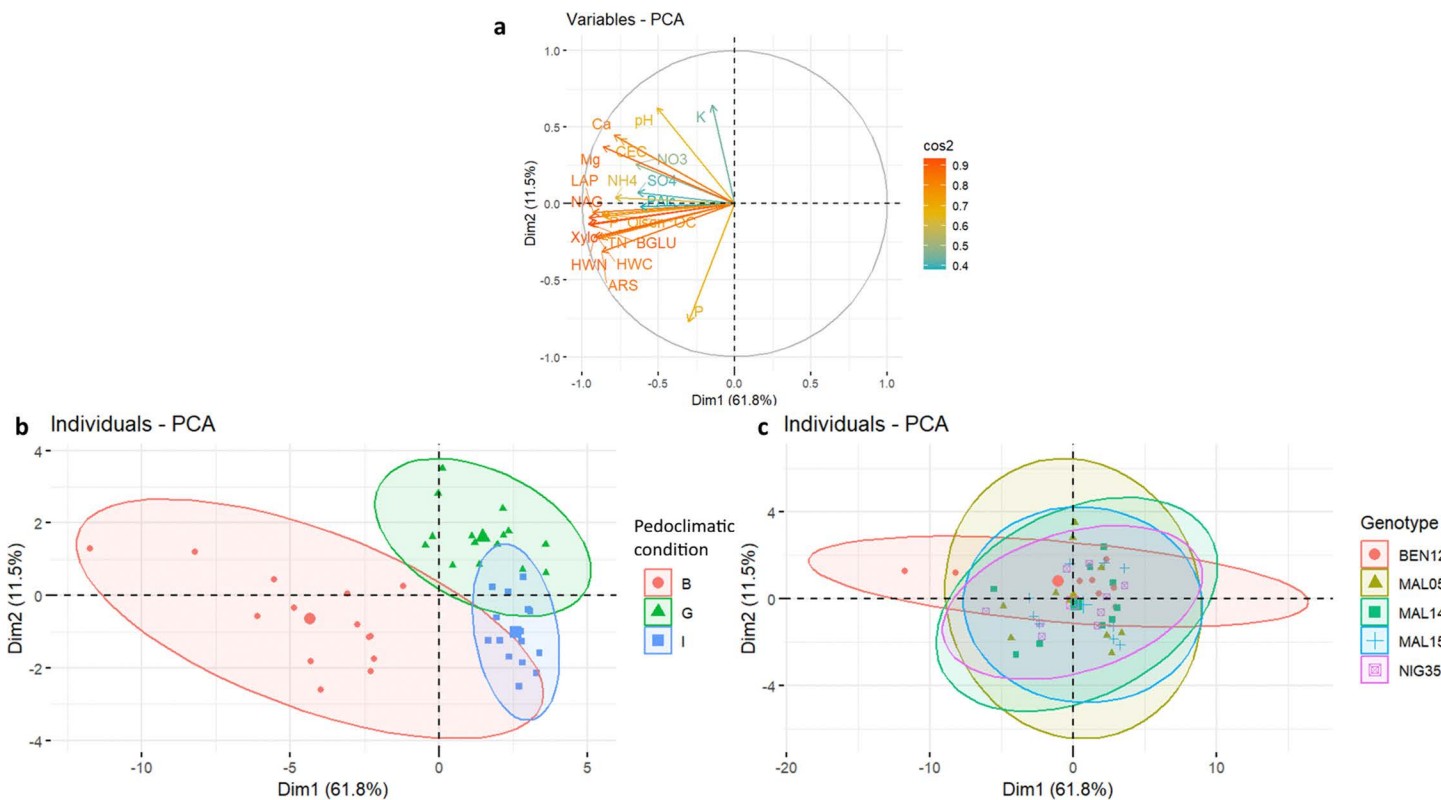

**Fig 1. Principal component analysis on soil chemical parameters and enzyme activities in the studied soils.** Correlation circle of Principal Component Analysis on the first two Principal Components **(a)**, Projection of individuals on the first two Principal Components grouping soil chemical properties and enzyme activities by pedoclimatic condition **(b)** and Fonio Genotypes **(c)**. cos2 indicates the quality of the representation of variables on the principal components/axis. Ca: Calcium, CEC: Cation exchange capacity, HWC: Hot water carbon, HWN: Hot water nitrogen, K: Potassium, Mg: Magnesium, NH4: Ammonium, NO3: Nitrate, OC: Organic carbon, P_Olsen: Phosphorus, pH: Soil pH, S04: Sulfate, TN: Total nitrogen, ARS: Arylsulfatase, BGLU: β-glucosidase, LAP: Leucine aminopeptidase, NAG: β-N-acetylglucosaminidase, P: Acid phosphatase, Palc: Alkaline phosphatase and Xylo: Xylosidase.

all pedoclimatic conditions; this decreased significantly from Boukoumbe (0.39 ± 0.04 cmol + /kg) to Gogounou (0.29 ± 0.02 cmol + /kg) and Ina (0.17 ± 0.01 cmol + /kg). Nitrogen fractions ($NH_4^+$, $NO_3^-$) had a similar trend; Boukoumbe demonstrated much higher $NH_4^+$ (16.64 ± 3.31 mg/kg) and $NO_3^-$ (27.71 ± 3.88 mg/kg) levels than the other pedoclimatic conditions. Boukoumbe and Gogounou exhibited similar pH levels (6.26 and 6.23, respectively), while Ina is more acidic (5.64 ± 0.07). Sulfate and phosphorus content presented a similar trend among pedoclimatic conditions; a higher level was observed at Boukoumbe, and a lower level at Gogounou and Ina. Boukoumbe recorded significantly higher hot water carbon and nitrogen, total nitrogen, and organic carbon compared to the other pedoclimatic conditions. For most soil chemical properties, there is no significant difference among fonio genotypes (Fig 1c, Table 4). For $K^+$ content, BEN12 was significantly higher (41.45 cmol + /kg) compared to MAL14, MAL15, and NIG35, which were lower. $Mg^{2+}$ content showed slight genotypic variation, as BEN12 had a significantly higher concentration compared to NIG35, but the remaining genotypes (MAL05, MAL14, MAL15) showed no significant differences from either of these extremes (Table 4).

As shown in Table 5, soil enzyme activities varied significantly among pedoclimatic conditions (P < 0.001), whereas no significant effect of fonio genotype was detected (P > 0.05). Both alkaline and acid phosphatase activities were significantly higher at Ina and Boukoumbe compared to Gogounou. Similarly, ARS, BGLU, LAP, NAG, and xylosidase activities exhibited significantly greater values at Boukoumbe than at the other pedoclimatic conditions.

**Table 4. Variation of rhizosphere soil chemical properties based on pedoclimatic conditions and genotypes studied.**

| | Pedoclimatic conditions | | | Genotypes | | | | |
|---|---|---|---|---|---|---|---|---|
| | Boukoumbe | Gogounou | Ina | BEN12 | MAL05 | MAL14 | MAL15 | NIG35 |
| C_org | 1.48±0.08**a** | 0.61±0.07**b** | 0.36±0.02**c** | 0.76±0.23 | 0.95±0.19 | 0.78±0.17 | 0.84±0.21 | 0.67±0.13 |
| TN | 0.12±0.01**a** | 0.03±0.01**b** | 0.03±0.00**b** | 0.06±0.02 | 0.07±0.02 | 0.06±0.02 | 0.05±0.02 | 0.05±0.01 |
| HWC | 238.61±15.76**a** | 136.66±7.44**b** | 152.06±6.5**b** | 190.37±31. | 169.34±20.1 | 169.62±19.03 | 161.61±13.0 | 182.59±19.19 |
| HWN | 35.94±2.67**a** | 14.48±0.95**b** | 17.46±0.46**b** | 25.42±5.53 | 23.86±4.23 | 22.25±3.16 | 18.81±2.69 | 21.64±3.63 |
| $NH_4^+$ | 16.64±3.31**a** | 3.32±0.38**b** | 3.36±0.77**b** | 13.26±5.38**a** | 9.35±3.90**ab** | 5.13±1.67**b** | 6.50±2.39**b** | 4.24±0.91**b** |
| $NO_3^-$ | 27.71±3.88**a** | 17.29±2.93**b** | 12.30±1.58**b** | 23.81±6.59 | 19.91±3.55 | 21.67±3.23 | 14.90±5.22 | 14.77±1.74 |
| P_Olsen | 6.46±0.52**a** | 3.58±0.23**b** | 2.71±0.18**b** | 4.77±0.94 | 4.51±0.76 | 3.88±0.68 | 3.831±0.60 | 4.07±0.54 |
| $SO_4^{2-}$ | 3.97±0.65**a** | 1.66±0.27**b** | 2.02±0.39**b** | 3.71±1.06 | 1.82±0.39 | 3.53±0.80 | 1.93±0.41 | 1.91±0.35 |
| pH | 6.26±0.08**a** | 6.23±0.06**a** | 5.64±0.07**b** | 6.20±0.13 | 6.06±0.16 | 6.04±0.16 | 5.98±0.149 | 5.94±0.07 |
| CEC | 4.19±0.26**a** | 3.71±0.24**a** | 2.92±0.1**b** | 3.71±0.49 | 3.62±0.36 | 3.37±0.28 | 3.50±0.27 | 3.77±0.24 |
| $Ca^{2+}$ | 2.11±0.21**a** | 1.64±0.17**a** | 0.78±0.04**b** | 1.60±0.40 | 1.62±0.26 | 1.36±0.31 | 1.36±0.22 | 1.54±0.22 |
| $K^+$ | 0.07±0.01**b** | 0.1±0.01**a** | 0.08±0.00**b** | 0.11±0.01**a** | 0.07±0.01**b** | 0.08±0.01**b** | 0.07±0.01**b** | 0.09±0.01**ab** |
| $Mg^{2+}$ | 0.39±0.04**a** | 0.29±0.02**b** | 0.17±0.01**c** | 0.34±0.08**a** | 0.29±0.04**ab** | 0.23±0.03**b** | 0.25±0.03**ab** | 0.29±0.03**ab** |

Organic carbon (C_org. %). calcium ($Ca^{2+}$. cmol+/kg). cation exchange capacity (CEC. cmol+/kg). potassium ($K^+$. cmol+/kg). hot water carbon (HWC. mg/kg). hot water nitrogen (HWN. mg/kg). magnesium ($Mg^{2+}$. cmol+/kg). ammonium ($NH_4^+$. mg/kg). nitrate ($NO_3^-$. mg/kg). phosphorus (P_Olsen. mg/kg). soil pH (pH). sulfate ($SO4^{2-}$.mg/kg). and total nitrogen (TN. %). Values (mean±standard error) within a column followed by no letters or the same letter are not significantly different according to Tukey and Dunn tests at P≤0.05.

**Table 5. Variation of rhizosphere soil enzyme activities based on pedoclimatic conditions and genotypes studied.**

| | Pedoclimatic conditions | | | Genotypes | | | | |
|---|---|---|---|---|---|---|---|---|
| | Boukoumbe | Gogounou | Ina | BEN12 | MAL05 | MAL14 | MAL15 | NIG35 |
| ARS | 0.14±0.01**a** | 0.05±0**b** | 0.05±0**b** | 0.07±0.02 | 0.07±0.01 | 0.08±0.02 | 0.07±0.01 | 0.09±0.02 |
| BGLU | 0.29±0.02**a** | 0.12±0.01**b** | 0.09±0**b** | 0.17±0.05 | 0.15±0.03 | 0.15±0.03 | 0.16±0.03 | 0.18±0.04 |
| LAP | 0.37±0.02**a** | 0.16±0.02**b** | 0.10±0.02**c** | 0.23±0.07 | 0.20±0.04 | 0.20±0.05 | 0.18±0.04 | 0.25±0.04 |
| NAG | 0.02±0**a** | 0.01±0**b** | 0.01±0**b** | 0.01±0 | 0.01±0 | 0.01±0 | 0.01±0 | 0.01±0 |
| P | 0.73±0.05**a** | 0.36±0.02**b** | 0.74±0.05**a** | 0.61±0.07 | 0.58±0.1 | 0.61±0.09 | 0.59±0.07 | 0.67±0.09 |
| Palc | 0.81±0.04**a** | 0.72±0.03**ab** | 0.74±0.01**a** | 0.80±0.06 | 0.71±0.03 | 0.75±0.04 | 0.74±0.02 | 0.77±0.04 |
| Xylo | 0.07±0.01**a** | 0.02±0**b** | 0.02±0**b** | 0.04±0.01 | 0.03±0.01 | 0.03±0.01 | 0.03±0.01 | 0.04±0.01 |

ARS: Arylsulfatase (μmol/h/g). BGLU: β-glucosidase (μmol/h/g). LAP: Leucine aminopeptidase (μmol/h/g). NAG: β-N-acetylglucosaminidase (μmol/h/g). P: Acid phosphatase (μmol/h/g). Palc: Alkaline phosphatase (μmol/h/g). Xylo: Xylosidase (μmol/h/g). Values (mean±standard error) within a column followed by no letters or the same letter are not significantly different according to Tukey and Dunn tests at P≤0.05.

## Bacterial diversity in the fonio rhizosphere by genotype and pedoclimatic condition

The bacterial community in rhizosphere soil associated with fonio was assessed in the different samples through a metabarcoding sequencing of 16S rRNA gene (V4V5) regions. A total of 1,866,096 sequences with an average of 42,411.27 were obtained after reads assembling and filtering steps. The alpha-diversity indices were calculated and presented in Table 6. The results revealed that there are no significant differences among pedoclimatic conditions and genotypes for all alpha-diversity indices assessed (Observed ASV, Chao1 estimator, inverse Simpson index and Shannon index). Observed ASV ranged in average from 754.6 (Ina) to 788.67 (Gogounou) and from 760.30 (MAL14) to 791.44 (MAL05) for pedoclimatic conditions and genotypes, respectively.

The bacterial community structures of all samples across pedoclimatic conditions and genotypes were illustrated by a non-metric multidimensional scaling (NMDS) ordination plot (Fig 2) based on the beta-diverty index, the Bray-Curtis

**Table 6. Analysis of alpha-diversity indices.**

| | Pedoclimatic conditions | | | Genotypes | | | | |
|---|---|---|---|---|---|---|---|---|
| | **Boukoumbe** | **Gogounou** | **Ina** | **BEN12** | **MAL05** | **MAL14** | **MAL15** | **NIG35** |
| **Observed ASV** | 773.7±22.2a | 788.7±17.4a | 754.6±19.2a | 764.9±27.9a | 791.4±24.1a | 760±30.8a | 767.2±22.57a | 777.1±24.6a |
| **Chao1 estimator** | 869.7±21.3a | 885.9±17.3a | 844.7±17.6a | 874.2±22.2a | 875.6±24.9a | 849.8±28.7a | 862.2±26.4a | 872.67±22.7a |
| **Inverse Simpson** | 54.17±11.21a | 52.09±15.24a | 32.82±6.01a | 31.45±8.45a | 52.20±17.02a | 62.62±20.79a | 42.06±13.97a | 40.96±10.81a |
| **Shannon** | 5.05±0.14a | 4.93±0.17a | 4.86±0.16a | 4.76±0.24a | 5.06±0.22a | 5.09±0.19a | 4.88±0.19a | 4.92±0.18a |

Values (mean ± standard error) within a column followed by the same letter are not significantly different according to the Tukey test at $P \leq 0.05$.

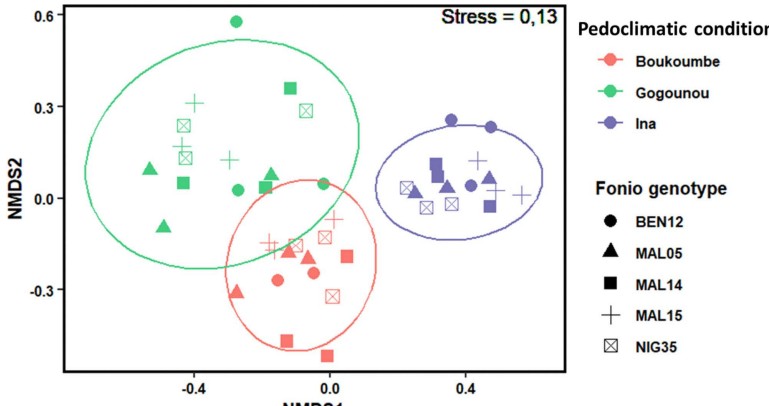

**Fig 2. Non-metric multidimensional scaling (NMDS) plot presenting the Bray-Curtis distance of the bacterial community by pedoclimatic conditions and fonio genotypes.**

distance. The stress value of 0.13 indicates an acceptable goodness of fit for the ordination, suggesting that the two-dimensional representation captured most of the variation observed. Distinct clustering was observed, showing a clear separation of the bacterial community structure across pedoclimatic conditions. The bacterial communities in Boukoumbe and Gogounou were clearly separated from the Ina bacterial community by the NMDS2 axis. For the fonio genotypes represented by symbols on the NMDS plot, the relative proximity of points within a cluster reflects similarities in bacterial community composition. These findings highlight the importance of spatial heterogeneity in shaping microbial community composition. According to a PERMANOVA applied on the beta-diversity (Table 7), the pedoclimatic condition explained 22% of the variation in bacterial communities, the genotype explained 7.8% and the interaction of pedoclimatic condition and genotype explained 15.3%. However, neither genotype nor the interaction had a significant influence on the diversity

**Table 7. Bacterial community structure variation explained by pedoclimatic condition and genotype (PERMANOVA on Bray–Curtis dissimilarities).**

| Variables | Bray-Curtis dissimilarities | |
|---|---|---|
| | **R² (%)** | **P (>F)** |
| **Pedoclimatic condition** | 21.99 | < 0.001 *** |
| **Genotype** | 7.89 | NS |
| **Pedoclimatic conditions * Genotype** | 15.28 | NS |

and community structure of fonio rhizosphere bacteria (P > 0.05). Only the pedoclimatic condition recorded a significant effect on the bacterial community structure (P < 0.001).

The analysis of relative abundance revealed 22 phyla, 44 classes, and 115 families (Fig 3). At the phylum level (Figs 3a and 3b), Proteobacteria and Bacteroidota were the most dominant, followed by Firmicutes, Actinobacteriota, Acidobacteriota, Cyanobacteria, and Verrucomicrobiota. The abundance of Proteobacteria and Bacteroidota changed significantly by the pedoclimatic conditions; Proteobacteria and Bacteroidota were, respectively, more abundant at Gogounou (52.5%) and at Boukoumbe (34.7%) than at other pedoclimatic conditions. Regarding the third most abundant bacterial community, Firmicutes, their relative abundance was significantly higher at Ina (9.3%) than at other pedoclimatic conditions (4.9% at Boukoumbe and 4.3% at Gogounou). Among the genotypes, the relative abundances of phyla were similar for all. Regarding the families of the Proteobacteria phylum, Moraxellaceae (38.20% in average) was the most abundant in all pedoclimatic conditions without significant variation. Oxalobacteraceae was significantly more abundant at Boukoumbe (20.3%) and Ina (22.2%) than at Gogounou (10.5%) (Fig 3c), while Comamonadaceae stood out in Gogounou (18.6%) with a high

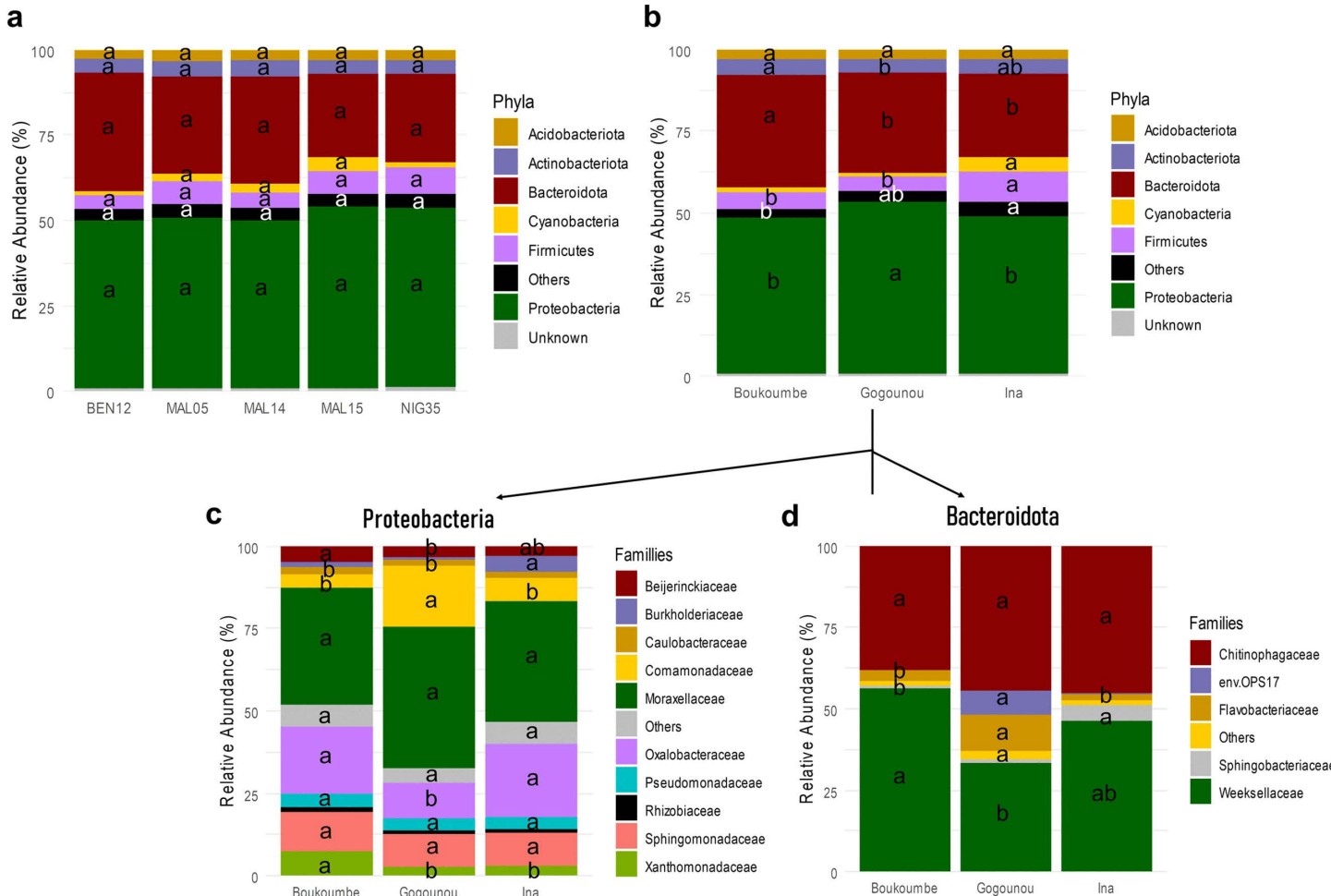

**Fig 3. The composition and relative abundance of major bacterial taxa in the rhizosphere soil of fonio.** The composition and relative abundance of major bacterial phyla with more than 1% relative abundance by genotypes **(a)** and by pedoclimatic conditions **(b)**. The composition and relative abundance of major bacterial families from the phylum Proteobacteria **(c)** and Bacteroidota **(d)**. «Others » represents groups with less than 1% relative abundance. Values designated by the same letters are not significantly different (Tukey HSD test; P ≤ 0.05).

abundance compared to the other two pedoclimatic conditions. We have also noted Sphingomonadaceae among the major families represented in the phylum Proteobacteria, with a similar abundance at all pedoclimatic conditions (10.6% on average). In the phylum Bacteroidota, the families Chitinophagaceae and Weeksellaceae were the most dominant (Fig 3d). Chitinophagaceae was equally abundant at all three pedoclimatic conditions (42.6% on average), while Weeksellaceae was more abundant at Boukoumbe (56.3%). Flavobacteriaceae, which was less represented, was found in large abundance at Gogounou (10.9%).

## Drivers of variation in rhizosphere bacterial community structure

Fig 4 presents redundancy analysis (RDA) plots performed to assess the soil drivers that modulate the bacterial community composition in the rhizosphere across the three pedoclimatic conditions studied. At the phylum level (Fig 4a), the first two axes explained 18.3% (RDA1) and 11.5% (RDA2) of the total variation, indicating the contribution of environmental factors in the rhizosphere bacteria of fonio. Key environmental variables among those studied here, pH, $NO_3^-$, and $SO_4^{2-}$ contents in soil, displayed strong directional arrows, suggesting their significant influence on the phyla distribution. The Bacteroidota phylum was closer to the soil pH, $NO_3^-$, and $SO_4^{2-}$ arrows, indicating a positive correlation between this phylum and drivers. The clustering of samples by pedoclimatic condition suggested site-specific bacterial community differentiation, with Boukoumbe and Gogounou showing overlaps, whereas Ina displayed a distinct composition; this confirms the pedoclimatic condition effect on bacterial communities identified. At the family level (Fig 4b), the RDA explained 24.5% (RDA1 and RDA2) of the total variance. The arrows representing environmental parameters suggested that pH, $NO_3^-$, and $SO_4^{2-}$ content in soil strongly influenced the bacterial composition. These parameters were positively associated with Weeksellaceae, Xanthomonadaceae, Sphingomonadaceae, and Chitinophagaceae families.

## Function prediction of bacterial community in fonio rhizosphere

A variety of functional traits of the fonio rhizosphere bacterial community were identified using Tax4Fun2 analysis. In total, 8,283 KEGG orthologs (KO) were detected across all samples and classified into six KEGG level 3 functional

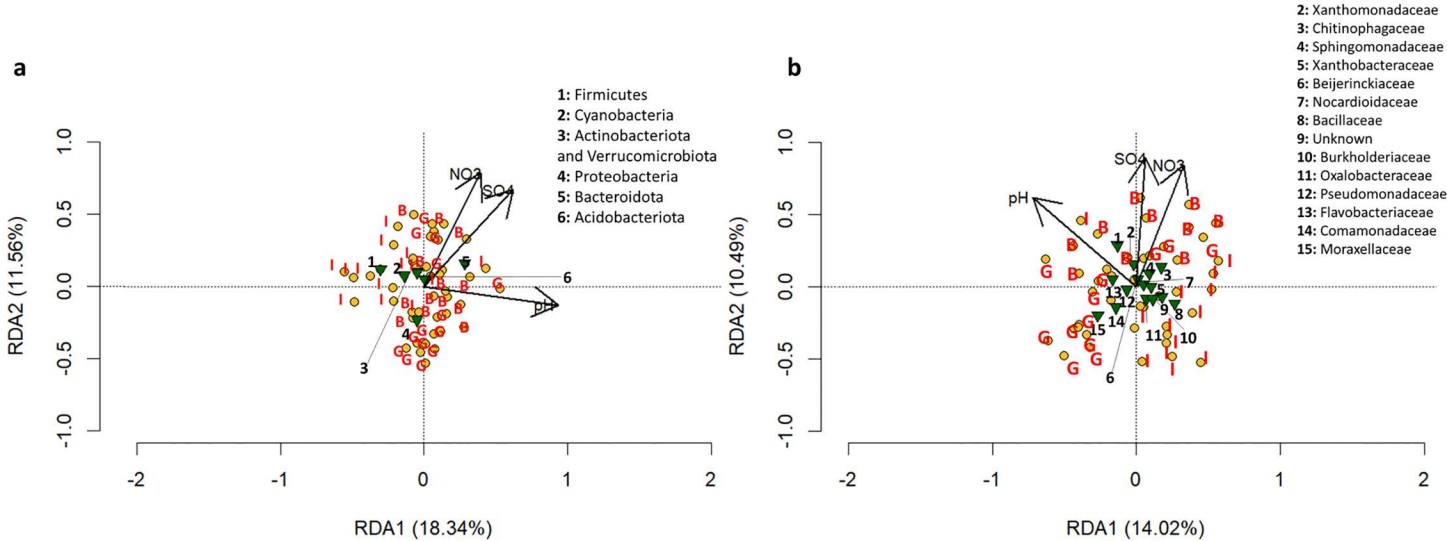

**Fig 4. Redundancy analysis (RDA) of the bacterial communities and the environmental parameters.** phyla **(a)**; the P values of each factor are P(pH) =0.001, P(NO3) =0.001, P(SO4) =0.024. family **(b)**; the P values of each factor are P(pH) =0.001, P(NO3) =0.001 and P(SO4) =0.056.Boukoumbe (B), Gogounou (G), Ina (I). Soil pH (pH), Nitrate (NO₃), and Sulfate (SO₄).

categories: metabolism (52.80%), environmental information processing (23.13%), cellular processes (13.08%), genetic information processing (6.79%), human diseases (3.25%), and organismal systems (0.95%). Among these, metabolism represented the most abundant functional module across the three studied pedoclimatic conditions. Pedoclimatic condition had no significant effect on metabolism, environmental information processing, and human diseases ($P > 0.05$) (S1 Table). In contrast, significant pedoclimatic condition effects were observed for cellular processes ($P = 0.03$), genetic information processing ($P < 0.01$), and organismal systems ($P < 0.001$) (S1 Table). Functional traits related to genetic information processing and organismal systems were particularly more abundant at the Boukoumbe site (Fig 5).

At the KEGG level 2, 46 KO groups were identified. Functions associated with global and overview maps were the most abundant (23.54%), followed by membrane transport (12.49%), signal transduction (10.64%), cellular community of prokaryotes (9.35%), carbohydrate metabolism (6.51%), and amino acid metabolism (5.15%) (Fig 6). Additional gene functions related to energy metabolism, lipid metabolism, xenobiotic degradation, nucleotide metabolism, terpenoid and polyketide metabolism, cofactor and vitamin metabolism, translation, replication and repair, cell motility, antimicrobial drug resistance, protein folding, sorting and degradation, cell growth and death, and glycan biosynthesis and metabolism were also detected. Significant differences among pedoclimatic conditions were observed for glycan biosynthesis and metabolism, cell growth and death, folding, sorting and degradation, metabolism of terpenoids and polyketides, xenobiotic biodegradation and metabolism, replication and repair, energy metabolism, carbohydrate metabolism, and cellular community of prokaryotes (S2 Table). Specifically, functions related to glycan biosynthesis and metabolism, folding, sorting and degradation, replication and repair, energy metabolism, carbohydrate metabolism, and cellular community of prokaryotes were significantly more abundant at the Boukoumbe site, whereas metabolism of terpenoids and polyketides, xenobiotic biodegradation and metabolism, and energy metabolism were more enriched at the Ina site.

## Discussion

### Rhizosphere soil biological (enzyme activities) and chemical properties

Soil biological and chemical parameters are crucial indicators of soil health and fertility, which condition agricultural productivity [53]. In this study conducted in northern Benin, the soil pH ranged from 5.3 to 6.75, indicating acidic conditions, confirmed by Amonmide et al. [37]. The soil chemical parameters show a significant variability across the pedoclimatic conditions, with generally low values reflecting the low fertility of the studied soils. Similar findings were reported by Hounkpatin et al. [4] about the CEC, the N and K content, in northern Benin using the Quantile Regression Forest (QRF)

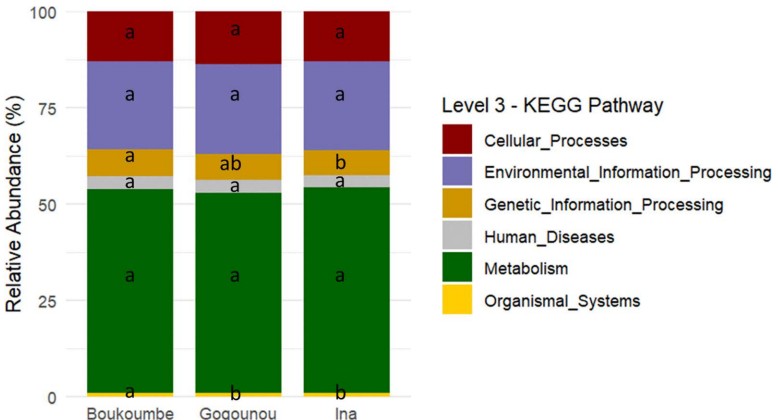

**Fig 5. Functional traits related to level 3 KEGG pathways in the fonio rhizosphere bacterial community based on Tax4Fun2 analysis.** Values designated by the same letters are not significantly different (Tukey HSD test; $P \le 0.05$).

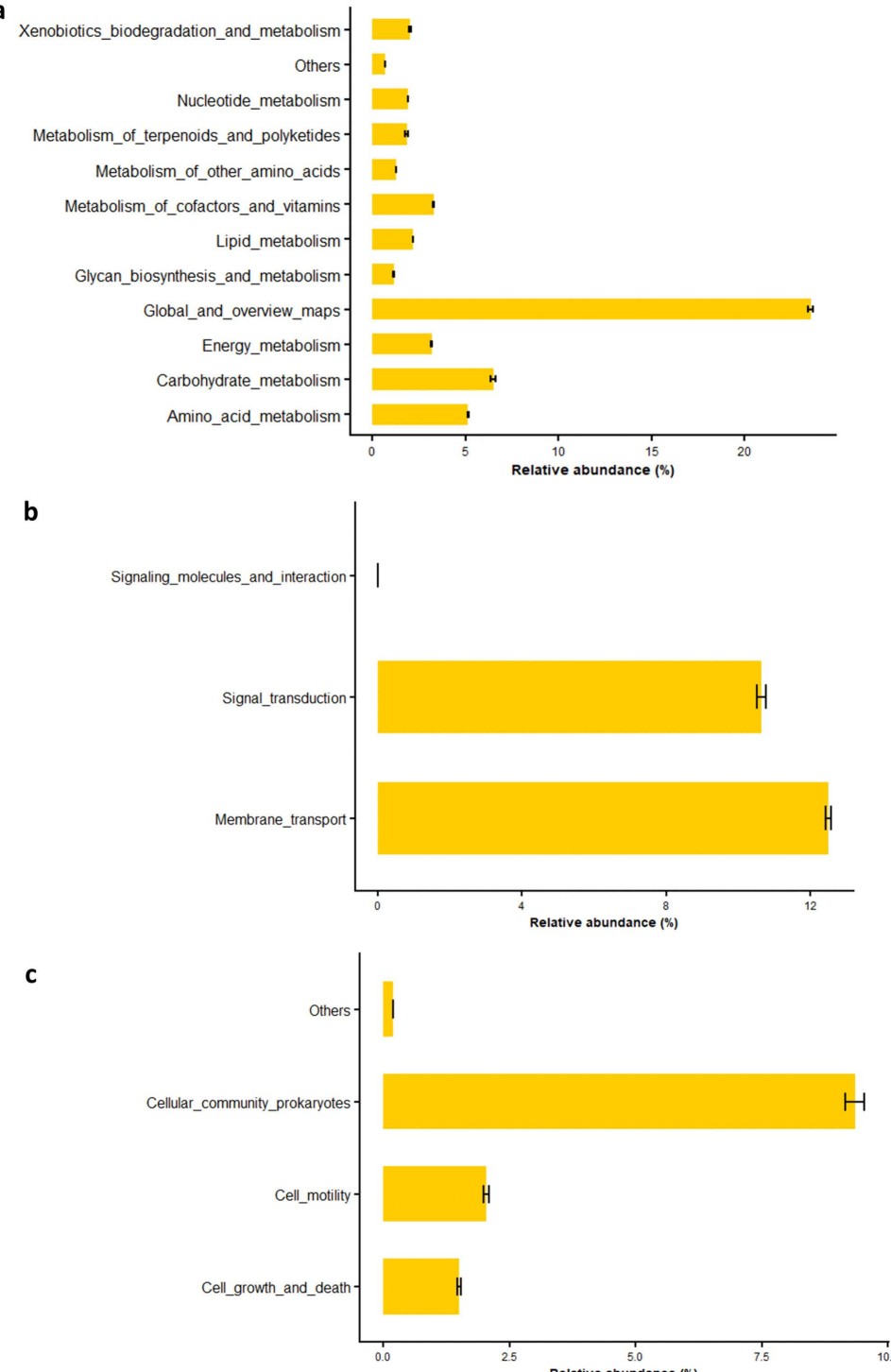

**Fig 6. Functional traits related to level 2 KEGG pathways in the fonio rhizosphere bacterial community based on Tax4Fun2 analysis.** Functions related to metabolism **(a)**, functions related to environmental information processing **(b)** and functions related to cellular processes **(c)**.

approach for soil properties mapping. Furthermore, according to the fertility classification of Sys [54], these soils belong to class IV, which corresponds to a low fertility level with a severe limitation of N, P, and K as well as a low organic matter content. The deficiency in nutrients is strongly correlated with the low organic carbon content, which is the main source of nutrients. This low organic carbon content results from agricultural practices such as burning, excessive use of chemical fertilizers, and removal of crop residues, commonly observed in these regions [55]. In addition, this kind of soil often has a very sandy physical aspect with low clay, resulting in poor water retention. The low clay content limits the nutrient retention capacity of the clay-humus complex, thus increasing the leaching of essential plant nutrients [37].

We recorded a higher enzyme activity in the Boukoumbe site than in the other sites. Soil enzymes are mainly produced by microbial communities for nutrition and directly degrade organic matter composed of larger compounds (chitin, hemi-cellulose, cellulose, sulfate or phosphate esters, etc) into simple organic and inorganic molecules [26,27]. Those studied here are involved in phosphorus, nitrogen, sulfur, and carbon cycling. The activities of soil enzymes in our study significantly varied across pedoclimatic conditions, with Ina presenting the lowest values. This could be attributed to a correlation between the soil enzymes and soil pH, since we also recorded low pH and enzyme activities at Ina site. Several authors have reported the impact of soil characteristics such as pH and organic matter availability, as well as environmental climate, on enzyme activities [15,26,56,57]. As in the study of Uwituze et al. [58], our results revealed a high positive correlation between NAG, the total N and soil pH. Indeed, NAG is a key indicator of chitin degradation, highlighting its role in nitrogen release [26,59]. Phosphatases (P and Palc) contribute to the mineralization of organic phosphorus and are more prevalent at all pedoclimatic conditions here [26,60]. This high activity is not linked to a high organic matter content, but to the limited phosphorus nutrition of microorganisms (bacteria and fungi). Phosphatases are secreted in large quantities when phosphorus levels are low, boosting the solubilization and mineralization of organic phosphorus [26]. Moreover, Waring et al. [61] reported that in tropical soils, particularly under acidic conditions, the NAG: P ratio tends to be low, indicating the higher microbial demand for phosphorus. Thus, the microbial community composition regulates the production of soil enzymes according to their nutrient requirements [57].

## Soil bacterial community structure, composition (abundance and diversity) and functions

Our results revealed that Proteobacteria and Bacteroidota were the most abundant phyla, followed by Actinobacteriota and Acidobacteriota in the fonio rhizosphere. Similar schemes have been reported in previous studies, with Mhete et al. [62] and Mushtaq et al. [63] confirming the high abundance of these phyla in semi-arid regions and millet rhizospheres. However, Ndour et al. [34] observed a dominance of Proteobacteria and Firmicutes in the rhizosphere soil of pearl millet. The most abundant phyla vary significantly among different millet species, as pointed out by Tian et al. [64] for proso millet, foxtail millet and sorghum. This is attributed to differences in root architecture and the specific exudates released by each crop root, which shape distinct bacterial communities [65–67]. Proteobacteria and Bacteroidota, representing more than 78% of the total bacterial communities in this study, are copiotrophic groups that play a key role in the decomposition of organic matter and nutrient cycling. Proteobacteria, gram-negative bacteria, are the most dominant phylum in the majority of soils [68], and their composition is influenced by factors, including genotype, pedoclimatic condition, and land use [30,69]. As observed here, Bacteroidota are generally more abundant in agricultural soils [70]. Their abundance varies according to cropping practices and seasonal changes [71], which explains the pedoclimatic condition effect on their distribution here. Additionally, we observed a higher abundance of Firmicutes and Cyanobacteria in the bacterial communities compared to other studies on millet and cereals. This could suggest a specific preference of these phyla for the fonio rhizosphere. For example, it has been reported that Firmicutes are more present in very acidic soils [72], as in the case of Ina in our study, which has the highest abundance of Firmicutes and the most acidic soil. But Ling et al. [73] reported the highest abundance of Firmicutes in Gramineae crops. In addition to these interesting characteristics of the community observed in this study, certain novel microbial strains found in millet rhizosphere have been reported to contribute to the degradation of pesticides [74].

These communities associated with the fonio rhizosphere may be involved in processes relevant to plant growth and soil functioning. Based on Tax4Fun2 analysis, six major functional modules were detected in the samples, with metabolism, environmental information processing, and cellular processes being predominant across all pedoclimatic conditions. Similar gene functions have also been reported by Adigoun et al. [75] in *Synsepalum dulcificum* roots and by Dong et al. [76] in cave environments. Moreover, Yang and Song [77]reported the predominance of metabolism-related functions during the degradation of solid wastes, where microbial communities are typically enriched in genes involved in nutrient transformation, carbon cycling, and energy metabolism. Such a high metabolic potential indicates that the fonio rhizosphere harbors an active bacterial community capable of supporting plant nutrition and soil fertility through organic matter degradation and nutrient mineralization. Although pedoclimatic condition had no significant influence on the abundance of metabolism, environmental information processing, and human disease-related functions, the significant effects observed on cellular processes, genetic information processing, and organismal systems suggest that abiotic factors, such as soil pH, play a role in shaping specific microbial functional traits [78]. The higher abundance of genes related to genetic information processing, and organismal systems at the Boukoumbe site may reflect enhanced microbial activity, consistent with the higher enzymatic activities recorded at this pedoclimatic condition. At KEGG level 2, the dominance of functions related to global and overview maps, membrane transport, and signal transduction suggests an active exchange of metabolites and signalling molecules within the rhizosphere. These processes are crucial for maintaining both plant-microorganism and microorganism-microorganism interactions, particularly in nutrient-limited soils [79]. Furthermore, the abundance of genes related to carbohydrate and amino acid metabolism in the rhizosphere supports the role of the bacterial community in decomposing root exudates and soil organic matter, thus contributing to carbon and nitrogen cycling [80].

## Bacterial communities shaped by pedoclimatic condition and edaphic characteristics

We hypothesized that the rhizosphere soil bacterial community is shaped by pedoclimatic conditions and fonio genotypes. This was not proven in our study; only pedoclimatic condition had a significant impact on the similarity of the bacterial community. Bacterial communities showed a high similarity among fonio genotypes, which contrasts with the results of Ndour et al [81] on pearl millet, where different lines significantly influenced bacterial community diversity, explained by the rhizodeposition [82]. The genotypes used in this study may exhibit similar root architecture and root exudate profiles, thereby reducing their influence on microbial community structure. This is consistent with the hot water extractable soluble carbon and nitrogen contents measured, used as indicators of rhizodeposition, which did not vary significantly among fonio genotypes. Several studies have also shown that soil microbial communities structure varies according to the plant developmental stage, with root exudates exerting a particularly strong influence during early growth phases [83]. In our study, sampling was conducted at a late developmental stage, which likely reduced the strength of exudate-driven selection effects. However, it has been established that variations in the microbial community caused by crop type are generally less important than those caused by pedoclimatic condition [64]. This is in agreement with our results, which showed that only 7% of the variation is explained by genotype, while over 21% is attributed to pedoclimatic condition. The strong influence of pedoclimatic condition has been widely reported and is probably related to differences in soil properties and cultivation practices specific to each region [14,20,21,64]. Numerous studies have highlighted the crucial role of edaphic characteristics in bacterial diversity in the rhizosphere of millet crops [64]. These characteristics determine bacterial development as a function of nutrient availability and tolerance to environmental stress. In this study, we observed that soil pH and nutrient content ($SO_4^{2-}$ and $NO_3^{-}$) are key factors in bacterial community distribution in the fonio rhizosphere. Among these, soil pH was identified as the main determinant of microbial community structure in various contexts: Muneer et al. [84] in red soils of pummelo orchards, Lammel et al. [85], Adigoun et al. [86] in Benin soils for *Synsepalum dulcificum*, and Tian et al. [64] for proso millet. Soil pH strongly influences the development of some microbial groups, and acidic soils present less bacterial diversity than neutral ones [22]. All the soils studied had an acidic pH, with the lowest values recorded in Ina. Soil acidity limits the availability of carbon for microorganisms, leading to microbial stress and limiting the

development of specific bacterial groups [87,88]. In addition, soil pH regulates the solubility of essential chemical elements required for bacterial nutrition, indirectly shaping the structure of the soil microbiome [85]. Another key factor influencing the distribution of the bacterial community in this study is the nitrate content of the soil. In the maize rhizosphere, Zhang et al. [89] reported that different forms of nitrogen, such as nitrate and ammonium, significantly modify the composition of the bacterial community. Nitrate availability tends to increase soil pH, thereby favoring the recruitment of bacterial taxa that thrive in higher pH conditions.

## Conclusion

Our study provides a comprehensive overview of the rhizosphere soil bacterial community associated with fonio across agroecological zones in northern Benin. We hypothesized that the chemical and microbial properties of fonio rhizosphere soils depend primarily on pedoclimatic conditions; this hypothesis is supported. We also expected that bacterial community structure would be shaped by both pedoclimatic conditions and plant genotype; this assumption is only partially validated. In fact, bacterial community structure in the fonio rhizosphere was influenced particularly by soil parameters, thus supporting our third hypothesis. Finally, we proposed that these communities would exhibit diverse functions related to soil health and fertility, and this hypothesis is confirmed. Together, these outcomes clarify the role of the rhizosphere microbiote and offer prospects for future research aimed at enhancing crop productivity.

## Supporting information

**S1 Table. Pedoclimatic condition effect on functions detected in the fonio rhizosphere bacterial community at KEGG level 3 pathways.**
(DOCX)

**S2 Table. Pedoclimatic condition effect on functions detected in the fonio rhizosphere bacterial community at KEGG level 2 pathways.**
(DOCX)

## Acknowledgments

Tania L. I. Akponikpè's PhD is a scholar of the University of Abomey-Calavi (Republic of Benin) and the Université de Lorraine (France). Her research work was funded by French National Research Agency through the PEA-BIOVALOR (ANR-21-PEA2–0006) Biovalor project. The funders had no role in study design, data collection and analysis, decision to publish, or preparation of the manuscript. We gratefully acknowledge Ms. Julie Genestier for her technical support and assistance with the training on all analyses of rhizosphere soils. We also thank PhD Andy Laprie and the LAE colleagues for the helpful scientific discussion. We thank the "Institut National des Recherches Agricoles du Bénin (INRAB)" and fonio farmers through "La Maison du Fonio" for giving their approval to implement our experimentation and collect the soil samples.

## Author contributions

**Conceptualization:** Elvire L. Sossa, Enoch G. Achigan-Dako, Guillaume L. Amadji, Séverine Piutti.

**Data curation:** Tania L. I. Akponikpè, Enoch G. Achigan-Dako, Séverine Piutti.

**Formal analysis:** Tania L. I. Akponikpè.

**Funding acquisition:** Enoch G. Achigan-Dako, Guillaume L. Amadji, Séverine Piutti.

**Investigation:** Tania L. I. Akponikpè.

**Methodology:** Tania L. I. Akponikpè, Séverine Piutti.

**Project administration:** Enoch G. Achigan-Dako, Séverine Piutti.

**Resources:** Enoch G. Achigan-Dako, Séverine Piutti.

**Software:** Tania L. I. Akponikpè.

**Supervision:** Elvire L. Sossa, Enoch G. Achigan-Dako, Guillaume L. Amadji, Séverine Piutti.

**Validation:** Séverine Piutti.

**Visualization:** Tania L. I. Akponikpè.

**Writing – original draft:** Tania L. I. Akponikpè.

**Writing – review & editing:** Elvire L. Sossa, Enoch G. Achigan-Dako, Séverine Piutti.

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
