## [Editor Report · Decision Letter 0]

27 Aug 2025

PONE-D-25-41886Diversity of soil bacterial communities in response to fonio genotypes and pedoclimatic conditions in BeninPLOS ONE

Dear Dr. AKPONIKPE,

Thank you for submitting your manuscript to PLOS ONE. After careful consideration, we feel that it has merit but does not fully meet PLOS ONE’s publication criteria as it currently stands. Therefore, we invite you to submit a revised version of the manuscript that addresses the points raised during the review process.

We look forward to receiving your revised manuscript.

Kind regards,

Massimiliano Cardinale, PhD

Academic Editor

PLOS ONE

Journal Requirements:

“This research was funded by French National Research Agency through the PEA-BIOVALOR (ANR-21-PEA2-0006) Biovalor project, within the framework of Tania L. I. Akponikpè’s PhD. “

**Additional Editor Comments:**

Academic Editor:

Fig. 2, Fig. 4 and Fig. 5 appears to have a low resolution (some labels in these figures are almost unreadable); this would hamper the possibility for Reviewers to peform appropriate reviews.

Please, re-sumbit a manuscript with high-resolution images with well-readable labels.

---

## [Author Response · Author response to Decision Letter 1]

8 Sep 2025

All answers are provided in our letter "Response to Reviewers".

---

## [Decision Letter · Decision Letter 1]

25 Sep 2025

PONE-D-25-41886R1Diversity of soil bacterial communities in response to fonio (Digitaria exilis Stapf) genotypes and pedoclimatic conditions in BeninPLOS ONE

Dear Dr. AKPONIKPE,

Thank you for submitting your manuscript to PLOS ONE. After careful consideration, we feel that it has merit but does not fully meet PLOS ONE’s publication criteria as it currently stands. Therefore, we invite you to submit a revised version of the manuscript that addresses the points raised during the review process.

We look forward to receiving your revised manuscript.

Kind regards,

Massimiliano Cardinale, PhD

Academic Editor

PLOS ONE

Journal Requirements:

Reviewers' comments:

Reviewer's Responses to Questions

**Comments to the Author**

1. If the authors have adequately addressed your comments raised in a previous round of review and you feel that this manuscript is now acceptable for publication, you may indicate that here to bypass the “Comments to the Author” section, enter your conflict of interest statement in the “Confidential to Editor” section, and submit your "Accept" recommendation.

Reviewer #1: All comments have been addressed

Reviewer #2: All comments have been addressed

Reviewer #3: (No Response)

2. Is the manuscript technically sound, and do the data support the conclusions?

Reviewer #1: Yes

Reviewer #2: Yes

Reviewer #3: Yes

3. Has the statistical analysis been performed appropriately and rigorously? 

Reviewer #1: Yes

Reviewer #2: Yes

Reviewer #3: Yes

4. Have the authors made all data underlying the findings in their manuscript fully available?

Reviewer #1: Yes

Reviewer #2: Yes

Reviewer #3: Yes

5. Is the manuscript presented in an intelligible fashion and written in standard English?

Reviewer #1: Yes

Reviewer #2: Yes

Reviewer #3: Yes

6. Review Comments to the Author

Reviewer #1: (No Response)

Reviewer #2: This revised manuscript addresses many editorial issues and presents a useful dataset showing that soil chemistry strongly structures the fonio rhizosphere bacterial community. The experimental approach (rhizosphere sampling, 16S sequencing, soil chemistry and enzyme assay) is appropriate, and the topic is of interest for neglected-crop microbiome research. The authors have responded to several reviews but still needs polishing.

1. Provide public accession(s) in a recognized repository lik NCBIand add the accession numbers

2. The choice to filter at 95% identity is unconventional for 16S ASV workflows. Authors must justify this threshold, clarify its impact on taxonomic resolution, and either reprocess with a stricter threshold or demonstrate that conclusions are unchanged.

3. why Archaea were removed (primer bias, negligible counts or what else) and provide summary counts showing they are negligible; otherwise include Archaea or report them separately.

4. Explicitly state the total number of sequenced samples and the sampling unit ,

5 Standardize taxonomic names (spellings such as Weeksellaceae, Actinobacteriota/Actinobacteria); ensure consistent taxonomy source .

6 Confirm and supply final high-resolution figure files (300–600 dpi.

7 Clarify enzyme-assay details (units, dry-weight basis, number of technical replicates).

8 Rephrase sentences to clearly distinguish alpha (no significant effect) versus beta (significant location effect) diversity results.

9 Consider functional inference (PICRUSt2/Tax4Fun) or correlation analyses between microbiome features and genotype performance (yield/height) to strengthen biological interpretation.

Best of luck

Reviewer #3: Overall Comments

The manuscript would benefit from (i) consistent use of 16S terminology, (ii) a brief justification of sequencing depth, and (iii) clearer reporting for enzyme activities (comparators, units, and tests). In places where biological interpretation relies on broad taxonomic ranks, adding genus-/strain-level anchors (or clearly flagging speculation) will strengthen the narrative.

Line-Specific Comments

Line 103:

Please use consistent terminology: 16S rRNA gene (i.e., 16S rDNA). Avoid forms such as “16S rDNA genes”.

Line 193:

How was the 11,057 reads/sample threshold determined? Please justify the cutoff (e.g., rarefaction curves, median depth, sensitivity to alternative thresholds) and list any samples excluded due to insufficient depth.

Lines 391–393:

“Higher/lower activity” is ambiguous. Higher/lower relative to which condition/site/timepoint? Specify assay conditions and units (substrate, temperature/pH, normalization basis), n and variability, effect sizes, and the statistical test/model (with multiple-testing correction, e.g., FDR). Avoid comparing magnitudes across different enzymes unless activities are placed on a common scale.

Line 405:

Do you mean high activity at all sites? Provide summary statistics (central tendency ± variability, n per site) and results of formal tests. If only some sites show elevated activity, qualify the statement accordingly.

Line 415:

Inferences at phylum/class (even family) level are too coarse to link relative abundance to drought resistance or other beneficial phenotypes. Either provide genus- or strain-level evidence (e.g., taxa with documented PGP traits, with citations) or clearly label the interpretation as speculative. Strengthen with effect sizes and FDR-adjusted p-values; the current text is qualitative.

Line 420:

“Composition of significant phyla” is unclear. If you mean phyla whose relative abundances differ significantly among groups, please state it explicitly and specify the taxonomic rank, statistical test, significance threshold, and multiple-testing correction.

7. PLOS authors have the option to publish the peer review history of their article (what does this mean? ). If published, this will include your full peer review and any attached files.

**Do you want your identity to be public for this peer review?** For information about this choice, including consent withdrawal, please see our Privacy Policy .

Reviewer #1: **Yes:** Ali Chenari Bouket

Reviewer #2: **Yes:** Saleem Ahmad

Reviewer #3: **Yes:** Mingfei Chen

---

## [Author Response · Author response to Decision Letter 2]

10 Nov 2025

All our answers are available in the reply letter.

---

## [Decision Letter · Decision Letter 2]

26 Nov 2025

PONE-D-25-41886R2Diversity of soil bacterial communities in response to fonio (Digitaria exilis Stapf) genotypes and pedoclimatic conditions in BeninPLOS ONE

Dear Dr. AKPONIKPE,

Thank you for submitting your manuscript to PLOS ONE. After careful consideration, we feel that it has merit but does not fully meet PLOS ONE’s publication criteria as it currently stands. Therefore, we invite you to submit a revised version of the manuscript that addresses the points raised during the review process. Please submit your revised manuscript by Jan 10 2026 11:59PM. If you will need more time than this to complete your revisions, please reply to this message or contact the journal office at plosone@plos.org . Please include the following items when submitting your revised manuscript:

We look forward to receiving your revised manuscript.

Kind regards,

Massimiliano Cardinale, PhD

Academic Editor

PLOS ONE

Journal Requirements:

Reviewers' comments:

Reviewer's Responses to Questions

**Comments to the Author**

1. If the authors have adequately addressed your comments raised in a previous round of review and you feel that this manuscript is now acceptable for publication, you may indicate that here to bypass the “Comments to the Author” section, enter your conflict of interest statement in the “Confidential to Editor” section, and submit your "Accept" recommendation.

Reviewer #2: (No Response)

Reviewer #3: All comments have been addressed

2. Is the manuscript technically sound, and do the data support the conclusions?

Reviewer #2: Yes

Reviewer #3: Yes

3. Has the statistical analysis been performed appropriately and rigorously? 

Reviewer #2: Yes

Reviewer #3: Yes

4. Have the authors made all data underlying the findings in their manuscript fully available?

Reviewer #2: Yes

Reviewer #3: Yes

5. Is the manuscript presented in an intelligible fashion and written in standard English?

Reviewer #2: Yes

Reviewer #3: Yes

6. Review Comments to the Author

Reviewer #2: several points require further justification to enhance the manuscript's impact.

The introduction excellently establishes the context of plant-microbe interactions and the importance of location vs. genotype. However, the specific rationale for hypothesizing a genotype effect in fonio is slightly underdeveloped.Briefly elaborate on why these specific agronomic traits (e.g., maturity cycle, plant height) might be expected to influence rhizosphere communities, perhaps via differences in root exudation patterns or growth dynamics, to better justify the genotype-related hypotheses.

Three replicated composite samples were collected per site..." This description is ambiguous. Does this mean there were 9 plants per genotype per location (3 plants x 3 composite samples)? Or were the three plants pooled to make one composite, and this was repeated three times (i.e., 9 plants total per genotype per location)?

In the Results (Fig 3b), it is stated that "Proteobacteria and Bacteroidota were more abundant at Gogounou (52.5%) and at Boukoumbe (34.7%) than at other locations, respectively." However, in the Discussion, the high abundance of Firmicutes at the most acidic site (Ina) is highlighted. The text would benefit from explicitly stating the actual measured abundance of Firmicutes at Ina compared to other sites to strengthen this observation.

The conclusion is strong but does not explicitly return to the four initial hypotheses to state which were supported and which were not.

To explain the lack of a genotype effect, the authors hypothesize that "the fonio genotypes studied here exhibit genetic similarity." This is a reasonable speculation, but no data on the genetic relatedness of these genotypes is provided. They originate from different countries (Benin, Mali, Niger), so they may not be genetically similar.

Reviewer #3: The authors have satisfactorily addressed all comments raised. I now recommend the manuscript be accepted for publication in its current form.

7. PLOS authors have the option to publish the peer review history of their article (what does this mean? ). If published, this will include your full peer review and any attached files.

**Do you want your identity to be public for this peer review?** For information about this choice, including consent withdrawal, please see our Privacy Policy .

Reviewer #2: **Yes:** Saleem Ahmad

Reviewer #3: **Yes:** Mingfei Chen

---

## [Author Response · Author response to Decision Letter 3]

7 Jan 2026

Thank you for your comments. We provide a point by point answer

---

## [Editor Report · Decision Letter 3]

8 Jan 2026

Diversity of soil bacterial communities in response to fonio (Digitaria exilis Stapf) genotypes and pedoclimatic conditions in Benin

PONE-D-25-41886R3

Dear Dr. AKPONIKPE,

We’re pleased to inform you that your manuscript has been judged scientifically suitable for publication and will be formally accepted for publication once it meets all outstanding technical requirements.

Kind regards,

Massimiliano Cardinale, PhD

Academic Editor

PLOS One
---

## [Editor Report · Acceptance letter]

PONE-D-25-41886R3

PLOS One

Dear Dr. AKPONIKPE,

I'm pleased to inform you that your manuscript has been deemed suitable for publication in PLOS One. Congratulations! Your manuscript is now being handed over to our production team.

Kind regards,

on behalf of

Dr. Massimiliano Cardinale

Academic Editor

PLOS One